# OpenReview forum: "One-shot Optimized Steering Vectors Mediate Safety-relevant Behaviors in LLMs"
_colmweb.org/COLM/2025/Conference — COLM 2025_

### Official Review · Reviewer_9bVR · 2025-05-10

**Rating:** 6
**Confidence:** 4
**Ethics Flag:** 1

**Summary:**

The paper proposes a novel method to learn steering vectors (SVs), learnable vectors that can be added, at inference time, to the model's layer activation to induce a particular behavior. The method improves existing approaches as it requires an individual prompt+answer string to learn each SV. Through an extensive experimental setup, the authors show that one-shot SV can effectively promote or suppress harmful behaviors, sometimes generalizing to different types of related inputs. Additionally, the authors relate SV intervention to "emergent misalignment", i.e., code models induced to generate malign code via SV also generate unsafe open-ended completions, and false information suppression, i.e., using SV to avoid that models self-correct when generating false facts.

Overall, the paper looks solid, with a valid motivation, research questions, and sufficient evidence to back up its claims.
While the tested models for each setup are limited, the experiments cover a wide range of behaviours and setups. The main weaknesses relate to density, clarity in writing, and the structuring of the manuscript.

**Questions To Authors:**

The authors recognize the high variance in the effectiveness of the learned SVs. Can this result be related to the length of the training examples? Since you are learning on a finite set of tokens, I would expect short inputs to provide smaller learning signals for the SV. In this sense, have you tried verifying the success rate vs. the length of prompt+response?

**Reasons To Accept:**

The paper expands on existing methods to learn SV and, more broadly, automatic behaviour steering. The proposed methodology has some variability, as the authors recognize, but follow-up research can mitigate it.

The results are consistent with the claims and research questions proposed.

**Reasons To Reject:**

The main weaknesses of the paper relate to information density and clarity in structuring. The manuscript becomes extremely dense from late Section 4 / early Section 5. Providing most of the experimental details in the appendix and the frequent lack of clarifying examples hinders readability and clarity. Places where I recommend including a clarifying example and/or details:
- L122-124, when introducing examples from the Posed testbed
- L141: The choice of layer 10 feels arbitrary. Is it something you repeated following prior work, validated through a validation set? Moreover, why learning five SVs specifically? Is this choice meant to reduce the overall variability? Or, in other words, how does the variability of behavior change when using more SVs?
- L185-190: The aspect of mode-connectivity is poorly introduced and unexpected, making it hard to relate to the authors' findings with some background introduction and preliminaries.
- L222: While relying on automatic LLM-based assessment is commonplace, I would avoid trusing it blindingly and add information on here on whether you manually validated the Llama's expected alignment outputs.
- L252-255: This passage is hard to follow and it remains unclear how to link the authors' findings to the model's internal causal mechanisms.
- L291-301: I recommend clarifying the behavior that you want to promote. It seems to me that promoting fake prompts aims to yield a model that continues generating false information and avoids backtracking. Either way, I would recommend flashing out what the goal of this steering is.
- L303: I suggest reporting one or two of such strings in the many body. Following the argument around the evaluation metric (i.e., the presence in the output of any such string) is hard without an example.
- L313-315: What is a fictitious/real "attribute" in this context? And what does it mean for it to be the primary one discussed in the generation?
- L319: You barely mention the (non-)reentrant SV optimization method, but then use it to discuss the results of "non-reentrant steered generations" not exhibiting FIR. It remains hard to understand what the main takeaway is here.

Having high variability on the SV effectiveness (as it is clearly shown in Fig 1, 2, and 3) hinders the generalizability of the proposed approach. In this sense, learning one-shot SVs is not very different from Mike & Turner's method, which requires searching for the best learned SV. I recommend discussing these aspects, parallelism, and limitations more in the introduction and conclusion of the work, as it is something that might prevent practitioners from relying on your method.

Since one-shot SV has been found to affect several setups, it is unexpected that the paper does not cover safety improvement, as tested by any state-of-the-art safety benchmark. Mitigating harmful responses is a primary motivation behind behavior steering and SV, but most tests in this paper evaluate the opposite task, i.e., inducing harmful responses in response to harmful instructions.

---

> ### Author Response · Authors · 2025-06-03
>
> Thank you very much for your detailed review, which we understand must have taken a while to compile. We were happy to see that you found that our work "looks solid", and that you appreciated its motivation and experimental evidence. It seems that your primary concerns with our work were with regard to a lack of clarity in the writing; we very much are grateful for your detailed suggestions on how to improve it, and will take them into account.
>
> We will now respond to your other concerns, and hope that we are able to adequately address them.
>
> > Having high variability on the SV effectiveness (as it is clearly shown in Fig 1, 2, and 3) hinders the generalizability of the proposed approach. In this sense, learning one-shot SVs is not very different from Mike & Turner's method, which requires searching for the best learned SV. I recommend discussing these aspects, parallelism, and limitations more in the introduction and conclusion of the work, as it is something that might prevent practitioners from relying on your method.
>
> Thank you for pointing this out; we will make sure to more clearly address this. However, we have also performed some additional experiments that find (weak) correlations between SV performance on the training example and SV effectiveness in general. In particular, for the refusal setting, we found a (weak) correlation ($r^2 \approx 0.29$) between the Harmbench classifier's output on the training example and its mean output on the test set. And in the Poser setting, there is a correlation between training loss and test set performance with $r^2 \approx 0.45$. For more details, see [here](https://anonymous.4open.science/r/one_shot_steering_supplemental_experiments-4A63/experiments.pdf). We believe that, while not perfect, these correlations demonstrate an advantage of targeted steering optimization over Mack and Turner's MELBO method: they suggest that when a steering target is present, then this target can be used to help guide the selection of well-generalizing steering vectors.
>
> > Since one-shot SV has been found to affect several setups, it is unexpected that the paper does not cover safety improvement, as tested by any state-of-the-art safety benchmark. Mitigating harmful responses is a primary motivation behind behavior steering and SV, but most tests in this paper evaluate the opposite task, i.e., inducing harmful responses in response to harmful instructions.
>
> Note that we do actually investigate a setting in which one-shot SV optimization is used to improve model safety: the Poser setting. In particular, our results show that one-shot SV optimization can improve safety on unknown prompts *even when we only train on prompts where the model behaves safely*. We believe that this setting reflects risks that might emerge in frontier models well: most models have been extensively safety-tuned, so if unsafe behavior emerges, it will occur on prompts that we do not expect to induce unsafe behavior. (For an example of this, consider [the incident where the safety-tuned Gemini production model suddenly sent death wishes to a user asking it for help with homework](https://www.cbsnews.com/news/google-ai-chatbot-threatening-message-human-please-die/).) This is why we do not test on standard safety benchmarks: because rather than attempt to make slight improvements to safety on prompts where instruction-tuned models are already very safe, we want to demonstrate that we can effectively modulate unsafe behaviors, so that we can obtain safety on unknown prompts as well.

---

> > ### Author Response · Authors · 2025-06-03
> >
> > We will now provide individual responses to each of your suggestions for improving clarity.
> >
> > * L122-124: We agree that providing an example of benign and harmful prompts for the Poser setting would be very useful, and will edit the manuscript to include this.
> > * L141: Layer 10 was chosen after testing steered model behavior at different layers on the Poser training set. (See Appendix C.) Note that we tested five SVs per norm to obtain more samples of model behavior when steered with different norms; this gives us a better estimate of the variability of behavior across different SVs.
> > * L185-190: The brief section on mode connectivity is largely an artifact of another line of research that we have been pursuing. We will remove it from the main body if it is too confusing.
> > * L222: We did validate the Llama misalignment/coherence evaluation outputs, and found that they mostly agreed with our assessment. We can add examples of these outputs in the appendix. (Note that when performing our validation, we found a minor bug in our evaluation code that led to coherence evaluations being paired with the wrong prompts. Fixing this bug did not qualitatively impact our results/conclusions at all, though, and we have updated our figures accordingly.)
> > * L252-255: To clarify: our main conclusion with regard to the FIR experiments is that it is possible for the steered model to stop outputting false information without having to explicitly output tokens that reflect this backtracking. This implies that outputting backtracking tokens like "Just kidding!" is not causally necessary for inducing backtracking---because if it were, then the steered model would continue to output false information. One way to think about this is in the context of work on unfaithful chain-of-thought, such as https://arxiv.org/pdf/2503.08679. We hope that this clarifies our conclusion.
> > * L291-301: You are correct in the goal of this steering in the FIR experiments; we will clarify this in the main body.
> > * L303: Thank you for pointing this out; we will include in this line the strings "Just kidding!" and "This is a joke" as examples of strings present in responses exhibiting FIR.
> > * L313-315: The fictitious attribute is an incorrect occupation associated with the entity, while the real attribute is the real occupation associated with the entity. We realize that referring to "attributes" instead of "occupations" is inconsistent, and will change this in the manuscript. Additionally, for an attribute to be the "primary" one discussed in the generation means that the model spends most of its response identifying the entity as having that occupation (even if it mentions another occupation), according to an LLM evaluator (see Appendix F.3.2).
> > * L319: The non-reentrant steering methods are the ones described in Section 2; reentrant steering is a more invasive steering method discussed in Appendix A. The intended takeaway here is that when attempting to steer the model to avoid outputting strings such as "Just kidding!", then instead of the model continuing to output fictitious information, it still reverts to outputting true information.
> >
> > In addition to your requests for clarification, we would like to also address your question:
> >
> > > Since you are learning on a finite set of tokens, I would expect short inputs to provide smaller learning signals for the SV. In this sense, have you tried verifying the success rate vs. the length of prompt+response?
> >
> > We found no correlation ($r^2 \approx 0.0093$) between prompt+response length and vector performance on Poser. For details, please refer to [this link](https://anonymous.4open.science/r/one_shot_steering_supplemental_experiments-4A63/experiments.pdf).
> >
> > ---
> >
> > We hope that our responses have clarified our work to you and addressed your concern. If you feel that we have adequately done so, then we would very much appreciate it if you would be willing to raise your score. Regardless, we would be happy to answer any follow-up questions that you may have. Again, thank you for all the time and effort that you put into your review.

---

> > > ### Comment · Reviewer_9bVR · 2025-06-06
> > >
> > > Thank you for your very detailed response. I get that most of the writing issues can be addressed in a camera-ready revision.
> > >
> > > > We found no correlation ($r^2 \approx 0.0093$) between prompt+response length and vector performance on Poser.
> > >
> > > This aspect remains somewhat surprising to me, as the number of output tokens determines the amount of learning feedback available for learning SVs. If not, what could be the reason behind the variability?
> > >
> > > At the same time, the weak correlation found between SV performance on training examples and downstream test performance still puzzles me -- how do we know that learned SVs generalize reliably?

---

### Official Review · Reviewer_KGN7 · 2025-05-12

**Rating:** 7
**Confidence:** 4
**Ethics Flag:** 1

**Summary:**

This paper introduces a new approach to finding steering vectors (SVs) in language models that can control safety-relevant behaviors. Unlike previous methods that require large contrastive datasets, the authors propose directly optimizing SVs through gradient descent on a single training example. The paper introduces several optimization techniques: promotion steering (maximizing probability of target outputs), suppression steering (minimizing probability of unwanted outputs), and mixed steering (combining both approaches). The authors evaluate their method across four diverse safety-relevant scenarios: (1) modulating harmful behavior in alignment-faking models, (2) bypassing refusal behaviors in instruction-tuned models, (3) investigating emergent misalignment when models are steered to produce vulnerable code, and (4) understanding how models handle the generation of fictitious information. Their results demonstrate that one-shot optimized SVs can effectively generalize across inputs, achieving high performance metrics such as over 95% attack success rate on Harmbench for refusal suppression. The paper makes a compelling empirical case that simple optimization on a single example may yield steering vectors that influence complex model behaviors across diverse inputs, providing both a tool for controlling model behavior and insights into the causal mechanisms underlying safety-relevant behaviors in language models.

**Questions To Authors:**

0. Layer selection in Poser experiments (pg. 14): Regarding your choice of layer 10 for SV optimization in the Poser experiments, could you elaborate on your preliminary evaluations? Specifically, how does performance vary across different layers, and is layer 10 consistently optimal across different types of safety-relevant behaviors?

1. Have you investigated the robustness of your one-shot optimized SVs to distribution shifts? For instance, would SVs optimized on one type of harmful prompt generalize to substantially different harmful prompts?

2. The mixed steering approach seems particularly promising. How does its performance compare to iterative approaches where promotion and suppression steering are applied sequentially?

3. The variance in SV performance is concerning. Have you explored regularization techniques that might make the optimization more robust across different training examples?

4. The results on FIR suggest a disconnect between a model's verbalized reasoning and its actual reasoning process. Do you see this as supportive or contradictory to mechanistic interpretability approaches that focus on analyzing explicit reasoning steps?

5.  After steering, how stable is the vector under continued finetuning or weight-diffusion techniques meant to remove jailbreaks? A small experiment would clarify defensive counter-measures.

6. In Poser, large-norm SVs behave best, but in refusal suppression norms above 10 sometimes degrade ASR. Have you explored adaptive norm scaling (e.g., line-search at inference) that practitioners could apply automatically?

----

*Bonus.* You might want to start the Appendix on a new page. Furthermore, Algorithm 1 on page 13 is misaligned.

**Reasons To Accept:**

This paper offers a compelling and timely contribution to the LM alignment and interpretability fields by demonstrating that one-shot optimized steering vectors (SVs)—those trained on a single example—can reliably mediate a wide range of safety-relevant behaviors across multiple models and settings. The core insight, that such vectors can be derived without requiring large contrastive datasets, is both practically valuable and conceptually intriguing. It lowers the barrier to steering model behavior and opens new paths for rapid red-teaming, behavior probing, and internal model diagnostics.

What sets the work apart is its breadth and depth of evaluation. The authors examine four distinct domains: revealing deceptive behavior in alignment-faking models, suppressing refusals in instruction-tuned LLMs, inducing emergent misalignment from a code-generation vector, and probing the mechanisms behind self-correction in the context of fictitious information. These are not trivial tests; they span classification, open-ended generation, code, and factuality. In each case, the SVs are not just effective in isolated cases but exhibit meaningful generalization across prompts and model inputs. Particularly striking is the Harmbench result, where a single SV yields a 96.9% attack success rate—a number that rivals or surpasses much more complex jailbreak strategies.

The methodology is carefully designed. The authors introduce and compare several steering objectives (promotion, suppression, mixed), test different vector norms and layers, and use appropriate controls (e.g., dot-product classifiers that fail even when the same SVs steer behavior). Their analysis is refreshingly honest, documenting variance across SVs and failure cases. The qualitative results—such as FIR being bypassed without loss of factual accuracy—also offer interesting clues about the causal structure of LLM reasoning.

In short, the paper is well-written, technically sound, and offers real empirical insights. It provides both a useful tool and a research probe into deeper alignment questions. While largely empirical, it is unusually thorough, and it makes a clear case for the practical and conceptual utility of single-example steering.

---
Overall, I am in favor of acceptance of this work at COLM.

**Reasons To Reject:**

The main limitation of the work is its lack of formal grounding. While the empirical findings are strong, there is little theoretical explanation for why single-example optimization works so well or when it might fail. This makes it difficult to assess generality beyond the tested models and tasks. Additionally, steering vector performance exhibits significant variance depending on prompt, model layer, and vector norm. For real-world application, some form of automated selection or reliability measure would be needed.

Moreover, the baselines are somewhat limited. For example, comparisons are missing against recent reinforcement learning or optimization-based jailbreak methods in the refusal setting. The safety risks of enabling high-success jailbreaks with simple SVs are also not deeply engaged with, which may raise concerns given the paper’s potential for misuse. Finally, the models tested are primarily open-source and mid-sized; whether the findings extend to frontier models remains uncertain.

---

> ### Author Response · Authors · 2025-06-03
>
> Thank you for taking the time to read our paper and to write your extremely detailed review. We are very happy to see that you appreciate the "practically valuable and conceptually intriguing" idea of one-shot steering vector optimization, along with the "impressive breadth and depth of evaluation" in our work. For the sake of brevity, we would like to note upfront that we performed additional experiments [here](https://anonymous.4open.science/r/one_shot_steering_supplemental_experiments-4A63/experiments.pdf) to address SV performance variability, which find that SV performance can be predicted from training example performance. Additionally, we wrote a theoretical model of one-shot steering vector optimization available [here](https://anonymous.4open.science/r/one_shot_steering_supplemental_experiments-4A63/theory.pdf), which we will add to the appendix of our manuscript.
>
> Because we found your concerns and questions to be very insightful, we would like to respond to them individually to the best of our ability. If you are interested, then you can find these responses below.
>
> > The main limitation of the work is its lack of formal grounding. While the empirical findings are strong, there is little theoretical explanation for why single-example optimization works so well or when it might fail.
>
> Although our paper is intended to be empirical, we have written up a formalization of one-shot steering vector optimization, along with a specific model that aims to provide theoretical intuition about a case in which one-shot steering vector optimization can be proven to work well. You can find our write-up at this link: [here](https://anonymous.4open.science/r/one_shot_steering_supplemental_experiments-4A63/theory.pdf).
>
> > Additionally, steering vector performance exhibits significant variance depending on prompt, model layer, and vector norm. For real-world application, some form of automated selection or reliability measure would be needed.
>
> In our current follow-up research, we are developing unsupervised methods for obtaining more reliable steering vectors from less effective ones. In the meantime, we have performed some additional experiments on the Poser and refusal settings, and found that steering vector performance on the single training example (weakly) predicts performance on test examples. In particular, for the refusal setting, we found a (weak) correlation ($r^2 \approx 0.29$) between the Harmbench classifier's output on the training example and its mean output on the test set. And in the Poser setting, there is a correlation between training loss and test set performance with $r^2 \approx 0.45$. This suggests that already, we can predict steering vector performance just from a single training example. (For more details, see [this link](https://anonymous.4open.science/r/one_shot_steering_supplemental_experiments-4A63/theory.pdf).)
>
> > Moreover, the baselines are somewhat limited. For example, comparisons are missing against recent reinforcement learning or optimization-based jailbreak methods in the refusal setting.
>
> Note that the approach of steering vector optimization differs from RL by requiring far fewer parameters to optimize, and differs from input-space-optimization methods like GCG by learning steering vectors in the hidden activation space of the model. That said, if you think that these comparisons would be particularly useful, we could potentially run experiments on applying GCG to a single training input.

---

> > ### Author Response · Authors · 2025-06-03
> >
> > We will now address your questions in order.
> >
> > 1. Our preliminary evaluations consisted of largely the same experimental procedure described in Section 3.2, with the difference that we only evaluated on the Poser training set. As described in Appendix C, other layers yielded steering vectors with qualitatively different behaviors (e.g. causing the model to output gibberish tokens when subtracting the vector, or flipping the model's behavior on both benign and harmful prompts when adding the vector). We do not necessarily believe that Layer 10 yields the highest-performing steering vectors across different safety-relevant tasks and different models---but much previous work in the literature has suggested that layers towards the middle of models tend to yield the best steering vectors.
> >
> > 2. In our Harmbench experiments, we trained each steering vector on a different type of harmful request, and found that the best steering vectors effected jailbreaks across all different types of prompts.
> >
> > 3. We have not yet performed ablation studies ourselves of mixed steering. Note that recent work on a steering optimization method called RePS (https://arxiv.org/pdf/2505.20809), which also uses a contrastive preference objective (like mixed steering), finds that it outperforms simple promotion steering (referred to as the language modeling loss in the RePS paper).
> >
> > 4. As mentioned earlier, we have found that SV generalization can be somewhat predicted from performance on the training example, which may allow the worst effects of SV performance variance to be mitigated. Currently, we are working on developing unsupervised methods for "amplifying" SV performance, which take a poorly-performing SV and yield a better-generalizing one. We have obtained some very preliminary positive results, but we still have to do more work before we are ready to share them.
> >
> > 5. To us, the FIR results imply that approaches to understanding and predicting model behavior by solely looking at explicit chain-of-thought reasoning are incomplete, and susceptible to being "fooled" by deceptive models. This is in line with recent results on unfaithful chain-of-thought such as https://arxiv.org/pdf/2503.08679.
> >
> > 6. We have not yet performed any such experiments. That said, the paper https://arxiv.org/pdf/2504.18872 found that performing latent adversarial training to remove jailbreaks both changes the model's representation of refusal (i.e. anti-refusal vectors obtained from the original model no longer induce jailbreaks) and yield new anti-refusal vectors that more effectively induce jailbreaks. (Do note that these vectors were not obtained using steering vector optimization as in our work.)
> >
> > 7. As a part of our current follow-up research, we are looking into developing unsupervised methods for choosing the SV norm at inference time. This is complicated by the desideratum that such a method not need to take into account any supervised information about what a desirable output would look like (since at that point, one could just use that information to optimize another steering vector), but nevertheless, we have a few promising approaches that interest us.
> >
> > (With regard to the bonus: duly noted.)
> >
> > ---
> >
> > Once again, we would like to thank you for your detailed response. We hope that we have adequately addressed your questions and concerns, and particularly would like to emphasize our new experiments on predicting SV generalization, along with our new theoretical formulation of one-shot SV optimization efficacy.

---

> > ### Comment · Reviewer_KGN7 · 2025-06-11
> >
> > I would like to thank the authors for their detailed and constructive response. They have thoroughly addressed my concerns. Overall, I am satisfied with their replies and support the acceptance of this work at COLM.

---

### Official Review · Reviewer_g4Zf · 2025-05-13

**Rating:** 7
**Confidence:** 3
**Ethics Flag:** 1

**Summary:**

This paper introduces a compelling and practical approach to optimizing steering vectors (SVs) in large language models. The authors systematically show that these one-shot SVs can modulate a wide range of safety-relevant behaviors—such as harmful response suppression, alignment faking detection, refusal bypassing, and behavior manipulation—across multiple models and evaluation settings.

Novelty: Direct SV optimization from a single input, avoiding reliance on large contrastive datasets.

Strong empirical results: Attack success rates over 90% in several setups; the ability to induce or suppress misalignment is both effective and interpretable.

Insightful analysis: The work contributes not only as a technique but also as a tool for understanding emergent and latent behaviors in LLMs.

**Reasons To Accept:**

The paper is well-written, thorough in experimentation, and offers both practical and conceptual value for the alignment, interpretability, and safety research communities.

**Reasons To Reject:**

I'm not an expert in this field, my evaluation might be wrong.

---

> ### Author Response · Authors · 2025-06-03
>
> Thank you for taking the time to review our paper. We are glad to hear that you appreciate our paper's novelty and empiricism, along with recognizing the ability that one-shot steering vector optimization has to contribute to obtaining deeper behaviors in LLMs. If you have any further questions about our paper, please do not hesitate to ask, and we would be happy to answer.

---

### Official Review · Reviewer_dKkm · 2025-05-13

**Rating:** 6
**Confidence:** 4
**Ethics Flag:** 1

**Summary:**

This paper introduces a novel method for controlling safety-relevant behaviors in LLMs by directly optimizing steering vectors on a single training example. It shows that one-shot SVs can generalize across diverse inputs to modulate behaviors like alignment faking, refusal suppression, emergent misalignment, and factual correction. The approach avoids reliance on large contrastive datasets and offers a more causally grounded mechanism for activation-level control. Through rigorous experiments across multiple models and tasks, the work provides strong evidence that SV optimization is both effective and interpretable, making a valuable contribution to LLM safety and alignment research.

**Reasons To Accept:**

The paper introduces a simple yet powerful technique for optimizing steering vectors (SVs) on just a single input, enabling behavior modulation without needing large contrastive datasets. This significantly lowers the barrier for intervention in safety-critical settings.

The one-shot SVs generalize well across diverse tasks, including alignment faking, refusal suppression, emergent misalignment, and factual correction, with results such as 93.4% success rate on harmful behavior induction and 96.9% attack success in HarmBench evaluations.

**Reasons To Reject:**

The method is used to induce or suppress behaviors but not to ensure safety or prevent misuse. Although valuable for probing model vulnerabilities, the paper does not demonstrate how this technique could be reliably used to improve safety at deployment time.

The success of SV generalization is shown empirically, but there is no underlying theory or mechanistic insight into why these one-shot vectors generalize so well across prompts and models.

---

> ### Author Response · Authors · 2025-06-03
>
> Thank you for taking the time to review our paper. We are happy to see that you appreciate the "rigorous experiments" in our work, and that you consider it to be a "valuable contribution to LLM safety and alignment research". We would like to briefly respond to your concerns, in the hope that we might be able to address them.
>
> > Although valuable for probing model vulnerabilities, the paper does not demonstrate how this technique could be reliably used to improve safety at deployment time.
>
> In our Poser experiments, we do show how one-shot steering could be used to improve model safety---particularly with respect to rare behaviors that cannot be easily elicited by normal means (such as simple prompting). What we do is find a steering vector that induces harmful behavior on a "normal" prompt on which the model behaves benignly, and then subtract that steering vector to cause the model to behave benignly even on harmful prompts. Importantly, the steering vector obtains this improvement in safety on an entire class of prompts that are unknown at training time (i.e. the "low oversight" prompts). This suggests that one-shot steering could be used to improve model safety at deployment time on unknown classes of prompts, even when the model behaves benignly on evaluations run before deployment time.
>
> > The success of SV generalization is shown empirically, but there is no underlying theory or mechanistic insight into why these one-shot vectors generalize so well across prompts and models.
>
> Thank you for pointing this out. While our paper was intended to primarily be empirical, we have written up a theoretical formalization of one-shot steering, along with a specific (simplified) model under which one-shot steering provably generalizes well. See [here](https://anonymous.4open.science/r/one_shot_steering_supplemental_experiments-4A63/) for more details.
>
> Again, thank you for your review, and for the time that you spent in writing it. We hope that we have been able to address your concerns; if so, then we would be grateful if you would consider raising your score.

---

> > ### Author Response · Authors · 2025-06-10
> >
> > Dear reviewer,
> >
> > As the discussion period is ending, we just wanted to follow up on our rebuttal addressing your concerns. We would appreciate if you could consider whether the clarifications that we provided warrant any adjustment to your evaluation scores, or if you have any additional questions or concerns we can address.
> >
> > Thank you!

---

### Decision · Program_Chairs · 2025-07-08

**Decision:**

Accept

**Comment:**

The paper introduces a method for controlling safety-relevant behaviors in LLMs by directly optimizing steering vectors (SVs) on a single training example. All reviewers agree that the core contribution - optimizing effective SVs from a single example - is novel and simple. However, the most significant concern, raised by dKkm, KGN7, and 9bVR, was that the strong empirical results lacked a theoretical explanation for why one-shot SVs generalize While the authors provided an anonymous link to the newly added theory, no reviewer commented on the quality of the added theory, and therefore, another round of detailed reviews is necessary to assess the accuracy of the theory. Another unresolved concern is around high performance variance across different SVs, which raised an important question of whether the learned SVs generalize reliably (9bVR). Overall, the paper requires substantial revision. However, the ideas are interesting and exciting. We urge the authors to improve the paper as much as possible given the reviewer comments, and to be upfront and clear about limitations (in case of issues that cannot be addressed and are fundamental).